# ON MEMORIZATION AND PRIVACY RISKS OF SHARPNESS AWARE MINIMIZATION

## ABSTRACT

In many recent works, there is an increased focus on designing algorithms that seek flatter optima for neural network loss optimization as there is empirical evidence that it leads to better generalization performance in many datasets. In this work, we dissect these performance gains through the lens of data memorization in overparameterized models. We define a new metric that helps us identify which data points specifically do algorithms seeking flatter optima do better when compared to vanilla SGD. We find that the generalization gains achieved by Sharpness Aware Minimization (SAM) are particularly pronounced for atypical data points, which necessitate memorization. This insight helps us unearth higher privacy risks associated with SAM, which we verify through exhaustive empirical evaluations. Finally, we propose mitigation strategies to achieve a more desirable accuracy vs privacy tradeoff.

## 1 INTRODUCTION

There have been considerable amount of recent works that explore loss optimization that searches for flatter optima (Norton & Royset, 2021; Foret et al., 2020; Wu et al., 2020; Kim et al., 2022; Du et al., 2022; Kwon et al., 2021). Flatness here measures how similar the loss value is for weight perturbations of certain degree around the optima. Significant empirical evidence has demonstrated that methods exploiting flatter optima tend to enjoy better generalization performance. While there have been works on explaining this improvement, these studies look at test accuracy as a monolith, and do not scrutinize on which specific test data points these performance gains come from, and what characterizes these points. In this work, our goal is to bridge this gap through the concept of *memorization*.

Overparamterized neural networks are powerful models capable of achieving close to zero training loss for many datasets. A key insight for this behavior stems from distinguishing 'learning' from 'memorization' (Feldman, 2020; Feldman & Zhang, 2020). Learning here refers to the classical process of *compressing* the training data into a model that is further used for predictive downstream task. Usually, such compression involves extracting and retaining pertinent information that is shared across groups of data points. For example, such groups may refer to data points sharing a class label, such as pictures of a tiger. The compression or learning task would be to delineate important features from the image that can help identify that the image is of a tiger. This could be made complicated by presence of *atypical* subgroups within groups, such as images of white tigers, which may not be as prevalent in the training data. In the worst case, there may only be a single image of a white tiger that is labeled as a tiger.

We can view deep neural network training as a combination of two tasks. The first task is that of relevant feature extraction or *representation learning*, which we can view as mapping the input space to a new space that is more amenable to the second task of learning classification boundaries. Presence of varying atypical subgroups, such as images of white tigers within images of regular tigers, can muddle up the task of relevant feature extraction for learning, especially if there are not enough number of such atypical data points in the training data. As has been observed by Feldman & Zhang (2020), there is indeed a long tail of such atypical subgroups within several benchmark image datasets.

Even if learning itself may be hard, due to overparameterization, a neural network can still achieve perfect training accuracy by *memorizing* such atypical groups. In the extreme cases, such as outliers

or other singleton data points that do not represent a (sub)group, non-generalizable features may be extracted and retained. For small subgroups, the training will involve discerning important subpatterns. The identification of such sub-patterns lies somewhere in between on the spectrum of perfect learning and perfect memorization. Indeed, even singleton images may have some generalizable features. For a test data point, one can estimate the effect of generalization impact of a training data point by its influence score (2) which approximates the change in prediction scores on the test data point if the training data point was removed from the training data before the model was trained.

Given the long tail of atypical examples in benchmark image datasets, and the corresponding significant improvement in generalization when using algorithms that seek wider minima, we show that there is a significant link between the two. This may seem counterintuitive, since Foret et al. (2020) show that seeking wider minima is robust against arbitrarily corrupted labels. However, the setup of Foret et al. (2020) shows only resistance to memorization of singleton data points, and does not preclude a weaker degree of memorization from subgroups. While some level of memorization may be important for good generalization, this can have unintended unfavorable consequences for privacy in machine learning. This is because higher memorization directly implies one could infer properties of training data from the model itself.

Protecting data privacy in deep learning models has gained considerable amount of interest in recent years. One aspect of privacy focuses on the question of whether certain data point(s) that were used to train the model could be reconstructed or be distinguished from other data that was not used for training. As a measure of privacy risk, many different attack settings have been developed including model extraction attacks (Carlini et al., 2021; Tople et al., 2020), attribute inference attacks (Fredrikson et al., 2015), property inference attacks (Ganju et al., 2018) and membership inference attacks (Shokri et al., 2021). Membership Inference (MI) Attack is one popular attack setup that we focus on in this paper. Given certain data, MI attack tries to predict whether the data was included in the original model's training data or not. The attack model typically takes the original model's output vector generated from the data of interest as additional information. A model can be said to have high membership privacy risk if the attack model can achieve high accuracy in classifying membership correctly. The core idea behind this attack is that there exists discernible differences between the output of data points that were in the training set and output of those that weren't. Sources of privacy risk have not been firmly established, but memorization and overfitting are some common intuitions offered in literature. Based on this, numerous defenses have been proposed to lower the privacy risk while keeping moderate test accuracy. Some works that have shown good trade-off between test accuracy and privacy risk involve instantiating an attack model and using adversarial methods or training multiple models with different partitions of the dataset and combining the models in some way. We focus specifically on unearthing privacy risk from the perspective of flat loss optimization.

Reflecting upon our finding that better generalization performance could come from the ability to generalize better on more memorized training points, we ask whether flatter minima induces higher privacy risk. Evaluation on 4 datasets indicates that this is true for a reasonably small weight perturbation ball around the minima. With this insight, we examine sharper minima as privacy defense mechanism as opposed to a flatter minima. While sharper minima has received relatively less attention in the literature, we discover that there is advantage to this optimization when it comes to protecting membership privacy. Combining this with the issue of over-fitting as source of privacy risk, we propose a new loss function, SharpReg, that exploits sharper minima while regularizing to prevent probabilities from going to extremes.

## 1.1 CONTRIBUTIONS

Our contributions are the following:

- We employ a novel methodology to dissect SAM's generalization gains, isolating the impact on data points that rely on varying levels of memorization. Towards this goal, we identify existence of a memorization spectrum via an influence entropy metric (Equation 3), with perfect learning on one end and perfect memorization on the other. This method can be applied universally to assess the generalization performance of any learning algorithm.

- We demonstrate higher membership privacy risk for SAM through extensive experiments as a caution to purely generalization-based search and adaptation of optimization algorithms. To the best of our knowledge, our work is the first to empirically explore an explicit link between an optimization algorithm and privacy.

- We propose mitigation strategies to achieve a more acceptable trade-off between generalization performance and privacy risk of SAM. Since our method is simple loss optimization, it can be efficiently trained and is highly extendable.

## 2 BACKGROUND & PRELIMINARIES

**Memorization & Influence scores**  For a training algorithm $\mathcal{A}$ which is trained using dataset $\mathcal{D} = ((x_1, y_1), ..., (x_n, y_n))$, the amount of label memorization by $\mathcal{A}$ on a sample $(x_i, y_i) \in \mathcal{D}$ is defined by Feldman (2020) by equation (1). Here, $\mathcal{A}(\mathcal{D}')$ is $\mathcal{D} - (x_i, y_i)$.

$$mem(\mathcal{A}, \mathcal{D}, i) := \Pr_{h \leftarrow \mathcal{A}(\mathcal{D})}[h(x_i) - y_i] - \Pr_{h \leftarrow \mathcal{A}(\mathcal{D}')}[h(x_i) - y_i] \tag{1}$$

In this paper, we will be following the definition given by Feldman & Zhang (2020) for estimating the influence of training example $(x_i, y_i)$ on test example $(x'_j, y'_j)$ using equation (2):

$$infl(\mathcal{A}, \mathcal{D}, i, j) = \Pr_{h \leftarrow \mathcal{A}(\mathcal{D})}[h(x'_j) - y'_j] - \Pr_{h \leftarrow \mathcal{A}(\mathcal{D}')}[h(x'_j) - y'_j] \tag{2}$$

**Sharpness Aware Minimization (SAM)**  Consider a model $f : X \rightarrow Y$ parameterized by a weight vector $w$ and a per-sample loss function $l: W \times X \times Y \rightarrow R_+$. Given a sample S = $\{(x_1, y_1),..., (x_n, y_n)\}$ sampled i.i.d. from a data distribution D, the training loss is defined as $L_S(w) = \sum_{i=1}^{n} l(y_i, f(x_i, w))/n$. Sharpness Aware Minimization combines traditional loss with sharpness term to minimize the difference between maximum loss in the vicinity (say a Ball of radius $\rho$: $B(\rho)$ ) of the current minima. Formally, it is defined as the following:

$$\min_w L_S(w) + [\max_{\epsilon \in B(\rho)} L_S(w + \epsilon) - L_S(w)]$$
$$= \min_w \max_{\epsilon \in B(\rho)} L_S(w + \epsilon)$$

There are many variants of SAM in the literature that enhance the idea proposed by SAM (Kwon et al., 2021; Abbas et al., 2022; Du et al., 2021; Zhong et al., 2022; Sun et al., 2023; Du et al., 2022).

### 2.1 MEMBERSHIP INFERENCE ATTACKS

Lets say that a classifier was learnt as f(x; $\theta$) from a training dataset $D_{train}$. We call this learnt model as the victim model V. Also, there is an attacker A who has access to an exact sample data point $x$ and the learnt model V. Under the definition of MI attack, A infers whether $x \in D_{train}$ or not. Based on the attacker's knowledge, there are many variations of the attack (Hu et al., 2022).

**Direct Single-query attacks**  The most commonly used MI attack is directly querying the target sample and using the statistics returned by the model to predict members and non-members of the training data with reasonable accuracy (Shokri et al., 2021; Murakonda & Shokri, 2020; Nasr et al., 2019; Zhang et al., 2021; Long et al., 2017; Sablayrolles et al., 2019; Yeom et al., 2020). Further details are in the Appendix A.2.

**Indirect Multi-query attacks**  Also known as the 'label-only attacks' because unlike the single query attack, the attacker can query multiple samples which are indirectly related to the target sample $x$ and use the predictions on these multiple queries to infer the membership of the sample $x$ (Hu et al., 2022; Li & Zhang, 2021; Long et al., 2018; Zhang et al., 2022). These multiple queries can extract additional information as a training sample influences the model prediction both on itself and other samples in its neighborhood. The main intuition behind the label-only attacks is that the model's accuracy and confidence in classifying samples near the member data should surpass its accuracy in classifying samples near the non-member data. In other words, members are expected to demonstrate greater robustness to any perturbation compared to non-members (Hu et al., 2022).

**Defenses against MI attacks**  There are many defenses which are explicitly designed to defend against MI attacks (Tang et al., 2022; Zheng et al., 2021; Nasr et al., 2018; Shejwalkar & Houmansadr, 2021; Huang et al., 2021; Jia et al., 2019) while other algorithms implicitly introduce

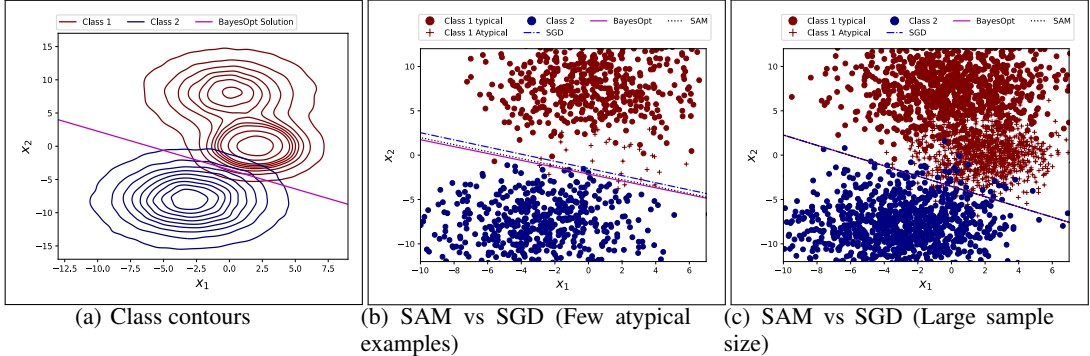

Figure 1: A toy construction illustrating the generalization ability of SAM over SGD for atypical examples. Fig (a) shows class density contours of a two-class, 2-dimensional classification problem, along with the Bayes Optimal solution. The red class has two 'clusters', one representing typical examples and one representing atypical examples. Fig (b) shows an instance of data sampled from densities shown in (a); the larger cluster of red dots represent typical examples in the red class, and the red '+' points represent a lot fewer atypical examples. SAM generalizes better than SGD in this case. Fig (c) shows that if there are enough samples generated from both typical and atypical clusters, SAM and SGD coincide with the Bayes Optimal classifier.

privacy against MI attacks like dropout , early stopping, label smoothening (Szegedy et al., 2016), Maximum Mean Discrepancy (Li & Zhang, 2021) and have been studied as defenses. Many explicitly designed methods make algorithmic changes, for example by adding noise as Differential Privacy Abadi et al. (2016) and MemGuard Jia et al. (2019), or use knowledge distillation based technique such as SELENA (Tang et al., 2022). Our goal in this paper is to focus on identifying memorization aspects of SAM and mitigating them. Our focus is not to compete with these other methods, albeit we conjecture that the sharpness-aware techniques we propose could be combined with these techniques to improve them.

## 3 MEMORIZATION AND FLATNESS OF OPTIMA

### 3.1 MOTIVATION: A TOY EXAMPLE

In this section, we provide a simple toy construction that illustrates how SAM can achieve better generalization performance vs vanilla SGD. The example is illustrated in Figure 1. The data is generated from two-dimensional densities illustrated in Figure 1(a). The densities are supported in two dimensions labelled as $x_1$ and $x_2$. There are two classes - the red class and the blue class. Figure 1(a) also shows the Bayes Optimal classifier. The red class has two 'clusters', one representing the typical examples (e.g. yellow tigers labelled as tigers), and the other representing the atypical examples (e.g. white tigers labelled as tigers). The data is sampled in such a way that we have several samples from the typical cluster, while there are only a few samples from the atypical cluster in the red class. This is shown in Figure 1(b). Figure 1(b) further shows that seeking flatter minima using the SAM optimizer learns a classifier that is closer to the Bayes Optimal classifier than the classifier learnt using vanilla SGD, and thus the former generalizes better. This difference in performance vanishes in Figure 1(c) when we have a large sample size for the atypical examples as well.

This toy construction shows that one possible reason that SAM and SAM-like algorithms can perform better is if they tend to memorize more than vanilla SGD. In other words, the gain in generalization could potentially come from those atypical data subgroups. In the next subsection, we empirically verify this conjecture for the CIFAR-100 dataset and SAM.

### 3.2 SAM'S GENERALIZATION GAIN AND MEMORIZATION

In this section, we aim to empirically dig deeper into the generalization performance gap between SAM and SGD by enumerating it at a finer granularity on the test data points, as opposed to just

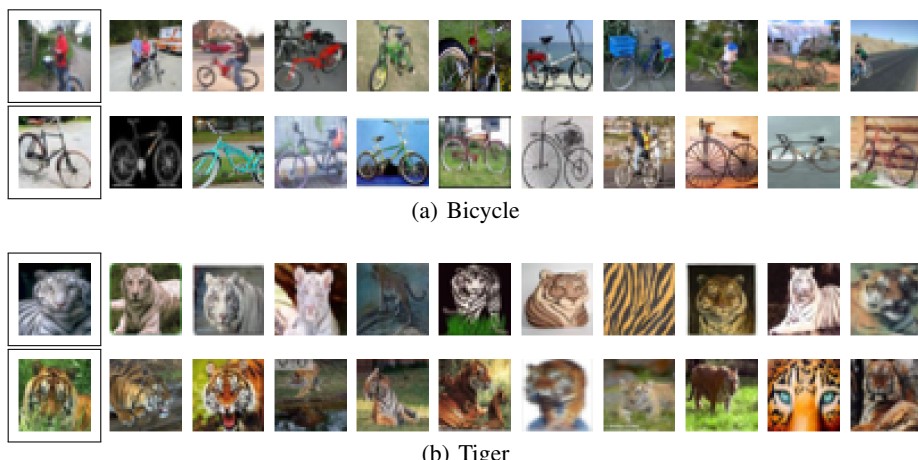

(a) Bicycle

(b) Tiger

Figure 2: Test images (boxed) from buckets 1 and 5 and their respective top-10 influential training images. For each object the top row is an image from bucket 1 and the bottom row is an image from bucket 5. For bucket 5 images (higher memorization,top row), notice that the images are atypical for their classes, and there is a near duplicate in the training data that was important for generalizing on this test image. For bucket 1 images, on the other hand, the top influential images are reminiscent of the test image at a conceptual level.

looking at the overall performance on the test set. We focus on the CIFAR-100 dataset. To do this, we use the influence scores calculated by Feldman & Zhang (2020) [1]. To evaluate the influence of any given training point on any particular test point for a model, one could remove that training data point, re-train the model and evaluate the difference in prediction probabilities on that test point. This is also called the Leave-One-Out (LOO) score given by equation (2). It is prohibitively expensive to evaluate these scores exhaustively for every training and test data point pair. Feldman & Zhang (2020) uses some clever sampling tricks to approximate these influence scores. They also calculate self-influence, or memorization scores for each training data point as a proxy for the change in prediction on a training point if it were to be removed from the training dataset. A higher memorization score for a training data point indicates a higher likelihood of it belonging to an atypical subgroup or even being a singleton in the worst case, and vice versa.

Our approach is to construct a metric that divides the test data points into groups based on the amount of memorization required for predicting them under traditional SGD learning. We then compare the performance on each group. We compare three learning algorithms on CIFAR-100 dataset: vanilla SGD, SAM, and SWA (Stochastic Weight Averaging (Wu et al., 2020)). SWA utilizes weight averaging across epochs while training. It implicitly seeks wider minima and achieves test performance similar to that of SAM. For SAM, we use $\rho = 0.1$ throughout the paper.

For each test point, we evaluate the entropy of the influence scores of the entire training points belonging to the same class as a representative value of how much memorization that test data point required. The idea is that test data points that require a great deal of memorization would be *greatly influenced* by only a few training data points (low entropy). Conversely, test data points that rely less on memorization would have influence scores more *evenly spread out* (high entropy) across training data points. For test data point $i$, let $\mathcal{S}_i$ be the set of all influence scores for training points of the same class (note that size of $\mathcal{S}_i$ is 500 for CIFAR-100 for example). Then, the entropy $\mathcal{I}_{ent}$ is calculated as:

$$\mathcal{I}_{ent}[i] = \sum_{i=1}^{n} -p_i * \log p_i, \text{ where, } p_i = \frac{|\mathcal{S}_i|}{\sum_i |\mathcal{S}_i|} \tag{3}$$

We group test data points into 5 buckets in the order of lowest $\mathcal{I}_{ent}$ to highest $\mathcal{I}_{ent}$. We present some test images and their top-10 influential training images in Figure 2 from bucket 1 and bucket 5. The

---

[1]We adopt precomputed influence scores and memorization scores on Cifar100 from https://pluskid.github.io/influence-memorization/

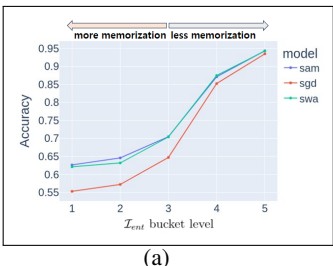 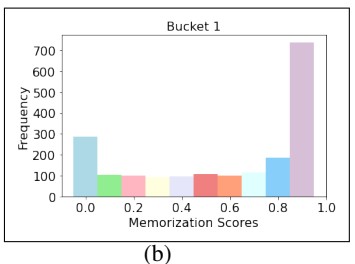 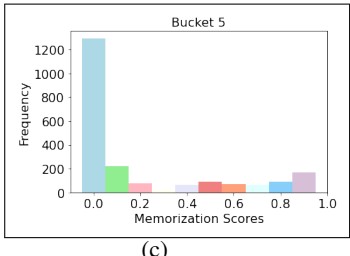

(a)            (b)            (c)

Figure 3: (a): Test accuracy on $\mathcal{I}_{ent}$ groups as evaluated by (3). (b) and (c): Distribution of top-1 memorization scores for bucket 1 and bucket 5.

figure illustrates that images from bucket 1 tend to be atypical images (e.g. bicycle alongside people, bottle held by a hand, white tiger) for their respective labels while images from bucket 5 tend to be more typical images (bicycle, bottle, orange tiger). For quantitative verification, we plot the distribution of memorization score of highest influencing training points for images from each bucket. Recall that memorization score is self-influence score for a training data point and approximates how much the model output would change for a specific training data point had the model been trained without that data point. We observe that lower numbered buckets are influenced by training points with higher memorization scores than higher numbered buckets (See Figures 3(b), 3(c)). The results for other buckets interpolate between those of bucket 1 and 5, and are skipped for brevity.

We compare the generalization gains of SAM and SWA against SGD on each of these buckets [2]. Figure 5(c) shows that for test data points in bucket 5, there is minimal performance gain, while for bucket 1, there is a more significant gain with other buckets interpolating in between. Thus, the performance gains of SAM/SWA can be attributed to more atypical data points which need more memorization. As such, one expects more privacy leaks from such models. Note that the buckets and memorization tendencies were calculated based on the models trained by SGD. It could be possible that SAM-like algorithms learn representations which encourage lesser memorization. But as observed by Feldman & Zhang (2020), and as also confirmed by our privacy leaks experiments, these memorization and influence scores are largely a property of the data, rather than that of model architectures or other variations in the training. Further, the connection between SAM's generalization and memorization also hints that if the dataset largely consists of the kind of data points that exist in bucket 5 (low memorization), then we may not see such generalization gains. Andriushchenko et al. (2023) suggested little or no correlation between flatter minima and generalization in their experiments when varying certain hyperparameters.

## 4   PRIVACY RISKS OF SAM

Based on the memorization trends discovered in Section 3.2 for optimization algorithms seeking wider minima, we examine whether SAM suffers from higher privacy risk by comparing the membership attack accuracy (refer to Section 2.1) on SAM and SGD trained models across four different benchmark datasets. We utilize target models that are widely employed in studies on membership inference attacks and defenses. Like other similar works, we assume that the attacker has access to some portion of the training data and non-training data that it uses to train the attack models.

**Datasets** We use CIFAR-10, CIFAR-100 and Purchase-100 and Texas-100. We follow Tang et al. (2022) to determine the partition between training and test data and to determine the subset that constitutes attacker's prior knowledge [3]. The details about the datasets can be found in Appendix B and about the experimental setup in Appendix C.

**Target Models** For CIFAR-100 and CIFAR-10, we use WideResNet (WRN) (Zagoruyko & Komodakis, 2016) with 16 layer depth and 8 as width factor. For Purchase-100 and Texas-100, we

---

[2] We do not use image transformations (e.g. random crop, rotations) to observe results without data augmentation. In later experiments we do use standard image transformations, but the trend is common both ways

[3] We adopt and extend the code in https://github.com/inspire-group/MIAdefenseSELENA

Table 1: Privacy vs Generalization tradeoff for SGD, SAM, and SharpReg (Higher query/label accuracy is worse for privacy).

| Dataset | Algo | Test Acc | Single-query Acc | Multi-query Acc |
|---|---|---|---|---|
| CIFAR-100 | SGD | 79.92% ($\pm$0.4%) | 76.68% ($\pm$0.38%) | 69.18% ($\pm$0.14%) |
| | SAM | 82.04% ($\pm$0.32%) | 79.09% ($\pm$0.56%) | 65.41% ($\pm$0.1%) |
| | SharpReg | 75.86% ($\pm$0.24%) | 59.64% ($\pm$0.69%) | 60.15% ($\pm$0.56%) |
| CIFAR-10 | SGD | 95.88% ($\pm$0.16%) | 59.05% ($\pm$0.3%) | 56.36% ($\pm$0.12%) |
| | SAM | 96.54% ($\pm$0.06%) | 61.32% ($\pm$0.35%) | 54.01% ($\pm$0.07%) |
| | SharpReg | 93.58% ($\pm$0.29%) | 53.30% ($\pm$0.64%) | 53.68% ($\pm$0.9%) |
| Purchase-100 | SGD | 84.95% ($\pm$0.38%) | 66.30% ($\pm$0.63%) | 65.27% ($\pm$0.33%) |
| | SAM | 84.83% ($\pm$0.4%) | 66.59% ($\pm$0.86%) | 65.84% ($\pm$0.24%) |
| | SharpReg | 81.34% ($\pm$0.61%) | 60.51% ($\pm$1.2%) | 60.64% ($\pm$1.07%) |
| Texas-100 | SGD | 50.67% ($\pm$0.4%) | 64.13% ($\pm$1.6%) | 63.61% ($\pm$1.5%) |
| | SAM | 51.50% ($\pm$0.24%) | 67.39% ($\pm$1.6%) | 66.27% ($\pm$1.6%) |
| | SharpReg | 49.78% ($\pm$0.72%) | 60.62% ($\pm$1.1%) | 59.27% ($\pm$0.69%) |

Table 2: Comparison of membership privacy and accuracy for SAM and SharpReg on SGD (% change)

| Dataset | Defense | Test acc diff | DSQ attack diff | Label-only attack diff |
|---|---|---|---|---|
| | SGD | +0.0% | +0.0% | +0.0% |
| CIFAR-100 | SAM | +2.65% | +3.14% | -5.45% |
| | **SharpReg** | -5.08% | -22.22% | -13.05% |
| CIFAR-10 | SAM | +0.69% | +3.84% | -4.17% |
| | **SharpReg** | -2.4% | -9.74% | -4.76% |
| Purchase-100 | SAM | -0.14% | +0.44% | +0.87% |
| | **SharpReg** | -4.25% | -8.73% | -7.09% |
| Texas-100 | SAM | +1.64% | +5.08% | +4.18% |
| | **SharpReg** | -1.76% | -5.47% | -6.82% |

follow the setting in Tang et al. (2022) and use a 4-layer fully connected neural network with layer sizes [1024, 512, 256, 100].

**Methods** We train models for sufficient number of epochs and choose the model with highest validation accuracy on a held-out validation set. We then employ different attack methods to evaluate the attack accuracy on the target model. We analyze the test accuracy and best attack accuracy values for direct single query attacks (DSQ) and multi query attacks. The details about the hyperparameters can be found in Appendix C.

**Results** The results are reported in Table 1. We report the mean and standard deviation over 5 randomized runs with different attack data splits. We observe that while SAM achieves higher generalization performance, it also demonstrates higher attack accuracy (i.e., higher tendency towards privacy leaks). This behavior is consistent across all the datasets that we test on. For further reliability we report consistent results using different architectures in Appendix E.

## 5 MITIGATING PRIVACY RISKS

In this section, we discuss techniques to mitigate the privacy risks associated with SAM.

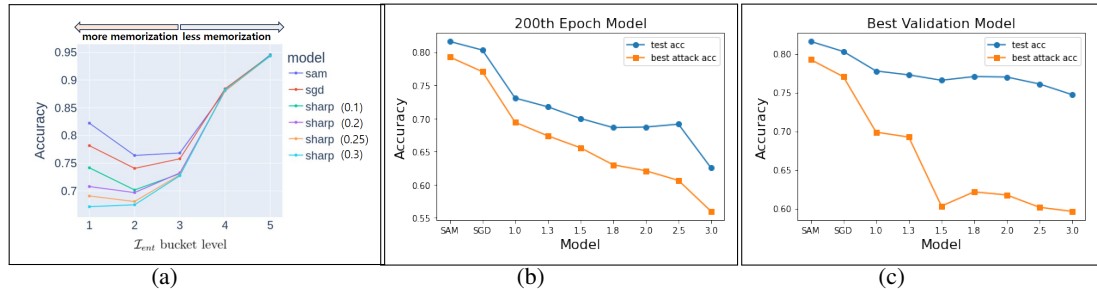

Figure 4: Tradeoffs of sharper minima by varying $\lambda$ as defined in Equation (4) for CIFAR-100. Higher (test) accuracy is better, lower attack accuracy is better. (a) Higher sharpness implies lower memorization (b) Higher sharpness implies lower generalization and lower privacy risk (c) Higher sharpness can be combined with early stopping to achieve a better generalization vs privacy tradeoff.

## 5.1 SHARPNESS VS PRIVACY TRADEOFF

If seeking flatter minima leads to more memorization (and hence, more privacy risks), what if we seek sharper minima instead? With this insight, we devise a new objective:

$$\min_w L(w) - \lambda[\max_{\epsilon \in B(\rho)} L(w + \epsilon) - L(w)] \tag{4}$$

Equation (4) modifies the loss minimization objective to seek minima that are sharper than what vanilla SGD would lead to. The sharpness itself is controlled by the hyperparameter $\lambda$, and higher values of $\lambda$ means sharper minima. Fig 4(a) illustrates how the generalization performance changes across the entropy buckets for CIFAR-100 for different values of $\lambda$ [4]. We see a clear trend – sharper minima tend to memorize less.

This further allows us to use the sharpness level $\lambda$ as a knob for generalization vs privacy tradeoff. This is illustrated in Figure 4(b) for CIFAR-100, where increasing lambda decreases test accuracy, but at the same time there is a corresponding decrease in privacy attack accuracy too. For this experiment, we ran the models to 200 epochs. However, we observed that if we use a validation set to select the best test accuracy, we can land on better attack accuracy, as illustrated in Figure 4(c). This reinforces the importance of early stopping, even for finding a good tradeoff between test accuracy and membership privacy risk. Early stopping hinders complete training on the training data, and achieves a better tradeoff. This implies that reaching 100 percent training accuracy is not ideal for our tradeoff. This could be explained as follows. We conjecture that each data point has generalizable and non-generalizable features. Clearly, only learning on the former will give better tradeoff. If training proceeds to learn the former features first, we should get better tradeoff which is what we observe. Motivated by this observation, we design another cost function that obviates early stopping by motivating a non-zero training-loss training.

## 5.2 PROPOSED NEW METHOD

We propose a new loss function, SharpReg, that searches for a sharper minima while motivating non-zero final training loss. We have the objective:

$$\min_w L(w) - \lambda[\max_{\epsilon \in B(\rho)} L(w + \epsilon) - L(w)] - \xi L(w) \tag{5}$$

$$= \min_w (1 + \lambda - \xi)L(w) - \lambda \max_{\epsilon \in B(\rho)} L(w + \epsilon) \tag{6}$$

The first term is the traditional training loss. The second term is the sharpness term. The third term controls for overfitting, and is a proxy to early stopping. When trained until zero training loss, the prediction vector is more prone to membership attack due to the extreme probabilities assigned to data points belonging to the training dataset. The third term assigns some mitigating effect in this

---

[4]For sharper minima, we found small $\rho$ value to be useful. We use $\rho = 0.01$ for our experiments

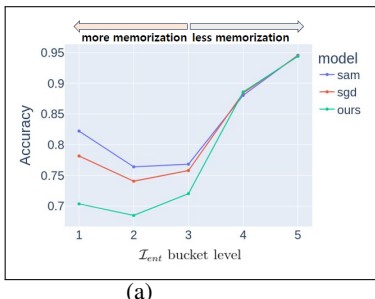 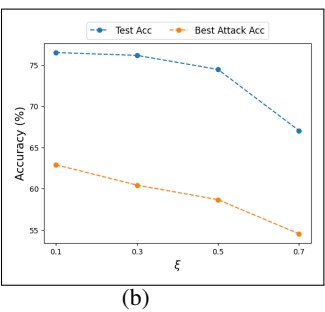 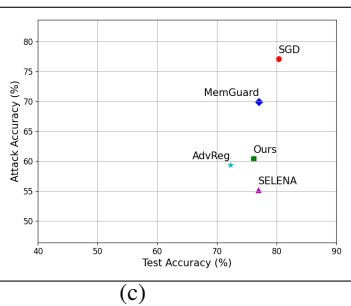

(a)           (b)           (c)

Figure 5: All experiments are on CIFAR-100. (a): Test accuracy on different $\mathcal{I}_{ent}$ levels for SAM, SGD, and SharpReg. (b): Change in test accuracy and attack accuracy with different values of $\xi$. (c): Comparison between various defenses

aspect despite some loss in generalization performance. This is illustrated in Figure 5(b) for fixed $(\rho = 0.01, \lambda = 1.5)$ (the behavior is similar for other values of $\lambda$). We note here, however, that model training can break if $\xi$ is too large and that adjusting $\xi$ cannot lower the privacy risk to arbitrarily low levels. We find it better to fine-tune $(\rho, \lambda)$ first and then adjust $\xi$ value for control over more fine-grained trade-off. We note that our loss function could be simplified to use 2 hyperparameters. We discuss this in more detail in Appendix C.4.1.

For our implementation, we adopt a straight-forward variant of the first order approximation method of Foret et al. (2020). We tune the hyperparamters to obtain a good tradeoff between memorization and test accuracy. Post-tuning with hyperparameters, $(\rho = 0.01, \lambda = 1.5, \xi = 0.3)$, the generalization on various entropy buckets is illustrated in Figure 5(a) which shows that the gap in test accuracy is larger for lower buckets with test data points that depend more on memorized training data points while high test accuracy is preserved for upper buckets with test data points that depend less on memorized training data points.

We report differences between test accuracy and attack accuracy for SAM and our method against traditional SGD optimization in Table 2. The values in the table indicates percentage increase or decrease when compared with SGD. For all the datasets, we consistently find that SAM achieves increase in test accuracy but also displays increase in membership inference attack accuracy. The proposed method finds reasonable tradeoffs on the test accuracy vs attack accuracy spectrum. e.g., for CIFAR-100, while test accuracy falls 5.08%, DSQ attack accuracy falls 22.22% and multi query attack accuracy falls 13.05%.

We compare our method with other standard privacy defenses - AdvReg (Nasr et al., 2018), Mem-Guard (Jia et al., 2019), and SELENA (Tang et al., 2022). Although explicit defense is not out goal and our method is based on only on simple optimization cost function modifications that does not require instantiation of attack models or training of multiple models on different partitions for self-distillation like some of these defense techniques, our loss optimization yields comparable tradeoffs. We illustrate our result for CIFAR-100 in Figure 5(c). Since our proposed modifications are purely based on altering optimization cost functions, combining them with other methods could lead to even better tradeoffs.

## 6   CONCLUSION AND FUTURE WORK

We have analyzed sharpness-aware minimization at a finer granularity level than before through the lens of memorization as a cautionary warning to adopting and adapting new optimization algorithms purely based on generalization performance. However, the proposed entropy metric is *post-hoc*. For CIFAR-100 that we analyzed, calculation of this metric required training of 4000 ResNet-50 models. While the insights we generate on memorization are useful, the proposed metric is not directly useful, which is why we have to settle for proxy cost functions that encourage less memorization. In future work, we hope to work towards identifying data points and features with less vs more memorization using more tractable approaches. Further, we would like to explore impact of sharpness-based SharpReg to privacy defense algorithms such as SELENA.

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
