# A ADDITIONAL RELATED WORKS

## A.1 CONNECTION OF FLATTER MINIMA WITH GENERALIZATION GAP

There have been numerous studies (Foret et al., 2020; Izmailov et al., 2018; Cha et al., 2021; Norton & Royset, 2021; Wu et al., 2020) which account for the worst-case empirical risks within neighborhoods in parameter space. Diametrical Risk Minimization (DRM) was first proposed by (Norton & Royset, 2021) and they asserted that the practical and theoretical performance of Empirical Risk Minimization (ERM) tends to suffer when dealing with loss functions that exhibit poor behavior characterized by large Lipschitz moduli and spurious sharp minimizers. They tackled this concern by employing DRM, which offers generalization bounds that are unaffected by Lipschitz moduli, applicable to both convex and non-convex problems. Another algorithm that improves generalization is Sharpness Aware Minimization (SAM) (Foret et al., 2020) which performs gradient descent while regularizing for the highest loss in the neighborhood of radius $\rho$ of the parameter space. (Izmailov et al., 2018) proposed Stochastic Weight Averaging (SWA) that performs averaging of weights with a cyclical or constant learning rate which leads to better generalization than conventional training. They also prove that the optima chosen by the single model is in fact a flatter minima than the SGD solution. Further, (Cha et al., 2021) argues that simply performing the Empirical Risk Minimization (ERM) is not enough to achieve at a good generalization, in particular, domain generalization. Hence, they introduce SWAD which seeks for flatter optima and hence, will generalize well across domain shifts.

## A.2 DIRECT SINGLE-QUERY ATTACKS

There are many variants of Direct Single-query attacks (DSQ) based on the approach of the attack and below we describe the ones used in our experiments:

**NN-based attack (Shokri et al., 2017; Tang et al., 2022; Nasr et al., 2018)** This is the first MI attack proposed by Shokri et al. (2017) where they use a binary classifier to distinguish between the training members and the non-members using the victim model's behavior on these data points. The adversary can utilize the prediction vectors from the target model and incorporate them along with the one-hot encoded ground truth labels as inputs. Then, they can construct a neural network ($I_{NN}$) called attack model.

**Confidence-based attack (Yeom et al., 2020; Salem et al., 2018; Song & Mittal, 2021)** If the highest prediction confidence of an input record exceeds a predetermined threshold, the adversary considers it a member; otherwise, it is inferred as a non-member. This approach is based on the understanding that the target model is trained to minimize prediction loss using its training data, implying that the maximum confidence score of a prediction vector for a training member should be near 1. The attack $I_{conf}()$ is defined as follows:

$$I_{conf}\hat{p}(y|x) = \mathbb{1}(\max \hat{p}(y|x) \geq \tau) \tag{7}$$

Here, $\mathbb{1}(.)$ is an indicator function which returns 1 if the predicate inside it holds True else the function evaluates to 0.

**Correctness-based attack (Yeom et al., 2020; 2018)** If an input record, denoted as x, is accurately predicted by the target model, the adversary concludes that it belongs to the member category. Otherwise, if the prediction is incorrect, the adversary infers that x is a non-member. This inference is guided by the understanding that the target model is primarily trained to achieve accurate predictions on its training data, which might not necessarily translate into reliable generalization when applied to test data. The attack $I_{corr}()$ is defined as follows:

$$I_{corr}(\hat{p}(y|x), y) = \mathbb{1}(\mathrm{argmax}\ \hat{p}(y|x) = y) \tag{8}$$

**Entropy-based attack (Nasr et al., 2019; Song & Mittal, 2021; Tang et al., 2022)** When the prediction entropy of an input record falls below a predetermined threshold, the adversary considers it a member. Conversely, if the prediction entropy exceeds the threshold, the adversary infers that the record is a non-member. This inference is based on the observation that there are notable disparities in the prediction entropy distributions between training and test data. Typically, the target model exhibits higher prediction entropy on its test data compared to its training data. The entropy of a prediction vector $p(\hat{y}|x)$ is defined as follows:

$$H(p(\hat{y}|x)) = -\sum_i (p_i log(p_i)) \tag{9}$$

where $p_i$ is the confidence score in $p(\hat{y}|x)$. Then, the attack $I_{entr}$ is given as:

$$I_{entr}(\hat{p}(y|x), y) = \mathbb{1}(H(p(\hat{y}|x)) \leq \tau) \tag{10}$$

**Modified entropy-based attack (Song & Mittal, 2021)**  Song et al.[15] introduced an enhanced prediction entropy metric that integrates both the entropy metric and the ground truth labels. The modified entropy metric tends to yield lower values for training samples compared to testing samples. To infer membership, either a class-dependent threshold $\tau_y$ or a class-independent threshold $\tau_{attack}$ is applied.

$$I_{Mentr}(\hat{p}(y|x), y) = \mathbb{1}(Mentr(p(\hat{y}|x)) \leq \tau_y) \tag{11}$$

where $Mentr(p(\hat{y}|x))$ for (x,y) data sample is given by combination of entropy information and ground truth label as:

$$Mentr(p(\hat{y}|x)) = -((1 - p(\hat{y}|x)_y)log(p(\hat{y}|x)_y) - \sum_{i \neq y}(p(\hat{y}|x)_i log(1 - p(\hat{y}|x)_i))) \tag{12}$$

## A.3  Label-only attacks (Multi-query attacks)

Also known as the 'multi-query attacks' because unlike the single query attack, the attacker can query multiple samples which are indirectly related to the target sample $x$ and use the predictions on these multiple queries to infer the membership of the sample $x$ (Hu et al., 2022; Li & Zhang, 2021; Long et al., 2018; Zhang et al., 2022). These multiple queries can extract additional information as a training sample influences the model prediction both on itself and other samples in its neighborhood. The main intuition behind the label-only attacks is that the model's accuracy and confidence in classifying samples near the member data should surpass its accuracy in classifying samples near the non-member data. In other words, members are expected to demonstrate greater robustness to any perturbation compared to non-members (Hu et al., 2022). Below we describe the multi-query attack setups used in our experiments, adopted from (Tang et al., 2022):

**Data augmentation attacks**  This attack was proposed by (Choquette-Choo et al., 2021) where the attacker generates additional data records by using augmentation methods like rotation, translation, adding noise etc. to the target image and query using these set of images, the membership is decided based on the correctness/confidence of the victim model on the set of these images. During the training process, we initially apply image padding and cropping, followed by horizontal flipping with a probability of 0.5, in order to augment the training set. Further, an attacker will query all potential augmented results of a target image sample. For instance, if the padding size for the left and right sides is 4 and the padding size for the top and bottom is also 4, and the size of the cropped image remains the same as the original image, there are (4 + 4 + 1) possible choices for left/right after cropping. Similarly, there are (4 + 4 + 1) possible choices for up/down after cropping. Additionally, considering horizontal flipping, there are 2 possible choices. Consequently, the total number of queries for a target image is 9x9x2 = 162. Given that the target model demonstrates a higher likelihood of correctly classifying the augmented samples of members compared to non-members, then, the target samples with a sufficient number of correctly classified queries will be recognized as members.

**Boundary estimation attacks**  *Boundary estimation attacks* (Li & Zhang, 2021; Choquette-Choo et al., 2021) is another type of label-only attack where the attacker can either introduce noise to identify adversarial examples that induce the smallest perturbation while altering the predicted label or utilize techniques for finding adversarial examples under the black-box assumption (Brendel et al., 2017; Chen et al., 2020). We use this attack as the label-only attack for our binary feature datasets - Purchase100 and Texas100. We introduce noise into the target sample by randomly flipping a specified number of features (Tang et al., 2022; Choquette-Choo et al., 2021; Li & Zhang, 2021). By setting a threshold on the number of flipped features, we generate numerous noisy samples per target sample for model querying. Subsequently, we conduct an attack by evaluating the percentage of correct predictions on the noisy samples to estimate the boundary. The intuition is that the samples located farther from the classification boundary are more likely to be correctly classified. Thus, the correctness percentage metric on the noisy samples can be employed to approximate the distance to the boundary.

## A.4  MI defenses

There are many defenses which are explicitly designed to defend against MI attacks (Tang et al., 2022; Zheng et al., 2021; Nasr et al., 2018; Shejwalkar & Houmansadr, 2021; Huang et al., 2021; Jia et al., 2019) while other algorithms implicitly introduce privacy against MI attacks like dropout , early stopping, label smoothening (Szegedy et al., 2016), Maximum Mean Discrepancy (Li & Zhang, 2021) and have been studied as defenses. *Differential Privacy* (Abadi et al., 2016) was studied in the context of Deep Learning for SGD optimization and is the only existing theoretical defense against all types of privacy attacks. The fundamental concept behind DP-SGD is to enhance privacy protection during model training by employing techniques such as clipping and adding noise to high gradients. This process helps to obfuscate the training data. There are some methods that perform confidence score masking to hide the true confidence scores of the target model. (Jia et al., 2019) proposes *MemGuard* which introduces a meticulously designed noise vector to the prediction vector and alters it to create an adversarial example for the attack model. On the other hand, (Nasr et al., 2018) utilized a min-max privacy game between the defense mechanism and the inference attack, to achieve privacy for the

defense model. Recently, some studies have focused their attention on knowledge distillation (Tang et al., 2022; Zheng et al., 2021; Shejwalkar & Houmansadr, 2021) to achieve significant privacy against MI attacks. (Tang et al., 2022) introduced SELENA which employs self-distillation to train a student model from multiple teacher models that were trained on different subsets of the data.

## B DATASETS

Here we introduce the four benchmark datasets used in the experiments and they have been widely used in prior works on MI attacks:

**CIFAR-10** [5] This is a benchmark dataset for image classification task. The dataset consists of 60,000 color images of 32x32 size. There are 6,000 images from 10 classes where 5,000 images per class belong to the training dataset and 1,000 images per class belong to the test dataset.

**CIFAR-100** [6] The dataset is designed to be more challenging than CIFAR-10 as it contains a greater number of classes and more fine-grained distinctions between objects. There are a total of 60,000 images from 100 classes. Each subclass consists of 600 images, and within each subclass, there are 500 training images and 100 testing images. This distribution ensures a balanced representation of each class in both the training and testing sets.

**Purchase-100** [7] This a 100 class classification task with 197,324 data samples and consists of 600 binary feature; each dimension corresponds to a product and its value states if corresponding customer purchased the product; the corresponding label represents the shopping habit of the customer. We use the pre-processed and simplified version provided by (Shokri et al., 2017) and used by (Tang et al., 2022).

**Texas-100** [8] This dataset is based on the Hospital Discharge Data public files with information about inpatients stays in several health facilities released by the Texas Department of State Health Services from 2006 to 2009. We used a prepossessed and simplified version of this dataset provided by (Shokri et al., 2017) and used by (Tang et al., 2022) which is composed of 67,330 data samples with 6,170 binary features. Each feature represents a patient's medical attribute like the external causes of injury, the diagnosis and other generic information. The classification task is to classify patients into 100 output classes which represent the main procedure that was performed on the patient.

## C EXPERIMENTAL SETUP

### C.1 $\mathcal{I}_{ent}$ EXPERIMENT

Here we discuss how test data points were grouped into 5 buckets according to different $\mathcal{I}_{ent}$ levels. Bucket 5 contains highest $\mathcal{I}_{ent}$ level, and is composed of test points where all 500 training points have 0 influence score. This means that the prediction output for that test point does not change had the model been trained without any one particular training data point. Because influence scores for all training points are equal, these test points have highest $\mathcal{I}_{ent}$ [9]. Figure 6(a) displays distribution of $\mathcal{I}_{ent}$ for remaining test data points. We group those above 6.1 into bucket 4. For the rest of the points, we calculate the mean and standard deviation and use them for grouping. We group points below $-0.4\sigma$ from the mean into bucket 1, points between $-0.4\sigma$ and $0.4\sigma$ into bucket 2, and points above $0.4\sigma$ into bucket 3. Final number of test points in each buckets are [Bucket 1: 1924, Bucket 2: 2996, Bucket 3: 2392, Bucket 4: 535, Bucket 5: 2153].

### C.2 ATTACK SETUP & SIZE OF DATA SPLITS

We adopt the attack setting from (Tang et al., 2022; Nasr et al., 2018) to determine the partition between training data and test data and to determine the subset of the training and test data that constitutes attacker's prior knowledge for CIFAR-100, Purchase-100 and Texas-100 datasets. We use similar strategy to determine the data split for CIFAR-10. Specifically, the attacker's knowledge corresponds to half of the training and test data, and the MIA success is evaluated over the remaining half. We report highest attack accuracy for multiple attack models in the main paper. Comprehensive results are discussed in D.

---

[5] https://www.cs.toronto.edu/ kriz/cifar.html

[6] https://www.cs.toronto.edu/ kriz/cifar.html

[7] https://www.kaggle.com/c/acquire-valued-shoppers-challenge

[8] https://www.dshs.texas.gov/THCIC/Hospitals/Download.shtm.

[9] When actually calculating $\mathcal{I}_{ent}$ with our formula (3), this evaluates to 0 due to normalization to probabilities, but represents highest value

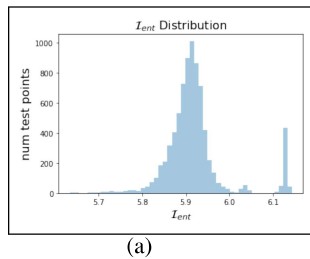

(a)

Figure 6: (a): $\mathcal{I}_{ent}$ distribution excluding bucket 5

## C.3 ATTACK ACCURACY

We delve into further detail on how we calculate the attack accuracy. Prior to training the attack model, we have already completed the training of the victim model V, which will be the target of the attack conducted by the attack model. During the training of model V, the dataset consists of two subsets, namely $D_{train}$ and $D_{test}$. In these subsets, the input feature corresponds to the image, while the label represents its true class.

While training the attack model, we further split $D_{train}$ into two equal halves, referred to as $D_{mem/train}$ and $D_{mem/test}$. Similarly, we split the $D_{test}$ into two parts, denoted as $D_{non-mem/train}$ and $D_{non-mem/test}$. The term "Mem" represents membership, indicating that the data was part of the training dataset for the victim model. Conversely, "Non-mem" denotes non-membership, signifying that the data was not included in the training dataset for the victim model. In these datasets, the feature set comprises of the image, its true class, and the output prediction vector obtained from model V. The associated label $y$ is a binary variable that indicates whether this particular data was part of the training data for model V. Now we define $D_{atr}$ and $D_{ate}$, which are training and testing dataset for the attack model. Denoting one data point as $d$ in equation (13),

$$D_{atr} = \{d | d \in D_{mem/train} \lor d \in D_{non-mem/train}\} \tag{13}$$
$$D_{ate} = \{d | d \in D_{mem/test} \lor d \in D_{non-mem/test}\} \tag{14}$$

Attack model learns a classifier from $D_{atr}$ and makes prediction for $i^{th}$ data point in $D_{ate}$,

$$\hat{y_i} = \arg\max_y p(y|x_i), y \in \{0, 1\} \tag{15}$$

The attack accuracy is then calculated as the percentage of correctly labeled input in $D_{ate}$. More formally,

$$A_{acc} = \frac{\sum_i^n \mathbb{1}(\hat{y_i} = y_i)}{n}, \text{where } |D_{ate}| = n \tag{16}$$

## C.4 COST FUNCTION AND HYPERPARAMETERS

In this section, we discuss simplification of our proposed cost function and describe the hyperparameters used to train the target models on each of the datasets.

### C.4.1 SIMPLIFYING OUR LOSS FUNCTION

Here, we note that our loss function (5) can be simplified to the following

$$\min_w L(w) - \beta \max_{\epsilon \in B(\rho)} L(w + \epsilon) \tag{17}$$

$$\text{where } \beta = \frac{\lambda}{1 + \lambda - \xi} \tag{18}$$

The simplified loss allows us to use two hyperparameters ($\rho,\beta$) as opposed to three. For our chosen hyperparameter values for CIFAR-100 ($\rho = 0.01, \lambda = 1.5, \xi = 0.3$), we can have equivalent loss with ($\rho = 0.01, \beta = 0.6818$). The loss function with three hyperparameters, however, allows for more intuitive understanding when fine-tuning the hyperparameters.

### C.4.2 BALL OF RADIUS $\rho$

For SAM loss, sharp minima loss, and our proposed loss, we approximate the maximum loss in the ball of radius $\rho$ around the minima. Norton & Royset (2021) have found that the type of norm that is used for defining the ball has large impact along with actual $\rho$ value. For all our experiments, we use L2 norm for our ball of radius $\rho$.

Table 3: Attack accuracy of different types of Direct Single-query attacks on SGD, SAM and our proposed method

| Dataset | Algo | NN | Confidence | Correctness | Entropy | Modified entropy |
|---------|------|-----|-----------|-------------|---------|------------------|
| CIFAR-100 | SGD | 76.86% | 59.71% | 77.04% | 76.70% | 76.97% |
| | SAM | 78.73% | 58.62% | 79.1% | 78.66% | 79.25% |
| | SharpReg | 57.62% | 58.42% | 59.69% | 57.88% | 59.69% |
| CIFAR-10 | SGD | 50.17% | 51.91% | 58.95% | 58.87% | 58.99% |
| | SAM | 50.01% | 61.12% | 51.64% | 61.10 | 61.81% |
| | SharpReg | 50.22% | 52.10% | 52.86% | 52.47% | 52.78% |
| Purchase-100 | SGD | 66.00% | 66.76% | 57.72% | 64.78% | 67.13% |
| | SAM | 66.62% | 67.30% | 57.53% | 65.35% | 67.54% |
| | SharpReg | 59.58% | 60.96% | 58.00% | 58.04% | 61.16% |
| Texas-100 | SGD | 59.81% | 65.20% | 63.30% | 55.74% | 65.13% |
| | SAM | 59.56% | 66.59% | 64.60% | 57.14% | 65.42% |
| | SharpReg | 51.11 % | 59.89% | 57.15% | 53.46% | 59.36% |

### C.4.3 HYPERPARAMETER TUNING FOR CIFAR-10 & CIFAR-100

We trained each model for 200 epochs and chose the model with highest validation accuracy on a held-out validation set. We used initial learning rate of 0.1 with learning rate decay of 0.2 at 60th, 120th, and 160th epoch with batch size of 128. We trained the models with weight decay 0.0005 and Nesterov momentum of 0.9. For SWA on CIFAR-100, we trained first 150 epoch with vanilla SGD and used weight averaging for the rest of the epochs.

We briefly discuss hyperparameter tuning. We fine-tuned hyperparameters more extensively for CIFAR-100 and adjusted similar values for other datasets. We first fine-tuned hyperparameters for sharp minima loss (4) in the order of $\rho$ then $\lambda$. With these values, we then fine-tuned $\xi$ for our proposed loss. For $\rho$, we tested values in [0.001, 0.005, 0.01, 0.05, 0.5, 1.0, 3.0]. We found that for sharp minima loss, small value of $\rho$ gives good tradeoff and chose 0.01 as our hyperparameter. For $\lambda$, we tested values in [0.01, 0.5, 1.0, 1.5, 2.0, 3.0]. We found that training breaks (training/test accuracy does not increase) for large value of $\lambda$. This may be because the sharpness term dominates the training objective. We chose 1.5 as a good $\lambda$ value for $\rho = 0.01$. Finally, we fine-tuned $\xi$ with [0.1, 0.3, 0.5, 0.7]. We do note that there is room for improvement with more hyperparameter tuning, and we leave this to future work.

### C.4.4 HYPERPARAMETER TUNING FOR TEXAS-100 & PURCHASE-100

We chose the best model as discussed before for CIFAR-10/100. We trained models with SAM, SGD, our proposed loss with a learning rate of 0.1 with weight decay 0.0005 and Nesterov momentum of 0.9. We trained the models on Purchase-100 for a total of 100 epochs and on Texas-100 for a total on 75 epochs. During training, we employed a batch size of 512 for the Purchase-100 dataset and a batch size of 128 for the Texas-100 dataset.

## D COMPREHENSIVE RESULTS FOR ALL ATTACKS

We report the test accuracy and MI attack accuracy values on all datasets and all methods for a single run in Table 4. For direct single query attacks, we evaluate attack accuracy for multiple attack methods explained above and report the highest attack accuracy.Additionally, direct single-query attack composes of multiple different attacks. We report attack accuracy of each attack algorithm in Table 3 for a single run. In the case of multi-query attacks, we conducted data augmentation attacks on computer vision datasets such as CIFAR-10 and CIFAR-100. Conversely, for binary feature datasets like Purchase100 and Texas100, we performed boundary estimation attacks and report their results in Table 4

## E COMPARISON OF DIFFERENT ARCHITECTURES

To validate consistency across different model architectures, we report results in Table 5 using InceptionV4 [10] and resnet18 [11] for CIFAR-100 and CIFAR-10. We kept our $\rho$ the same across all model architectures with

---

[10]https://github.com/weiaicunzai/pytorch-cifar100/blob/master/models/

[11]https://github.com/inspire-group/MIAdefenseSELENA/tree/main

Table 4: Comparison of membership privacy and accuracy on test/training set ($\lambda, \rho, \xi$)

| Dataset | Defense | Train acc | Test acc | Best Single Query | Best Multi Query |
|---|---|---|---|---|---|
| CIFAR-100 | SGD | 99.98% | 80.30% | 77.04% | 69.07% |
| | SAM | 99.98% | 81.6% | 79.25% | 65.45% |
| | MemGuard | 99.98% | 77.00% | 68.70% | 69.9% |
| | AdvReg | 89.39% | 72.24% | 58.39% | 59.29% |
| | SELENA | 80.31% | 76.92% | 55.15% | 53.68% |
| | **SharpReg(1.5,0.01,0.1)** | 96.39% | 76.48% | 62.15% | 62.90% |
| | **SharpReg(1.5,0.01,0.3)** | 93.21% | 76.14% | 59.69% | 60.44% |
| | **SharpReg(1.5,0.01,0.5)** | 89.56% | 74.44% | 58.42% | 58.68% |
| | **SharpReg(1.5,0.01,0.7)** | 76.42% | 67.04% | 54.34% | 54.55% |
| CIFAR-10 | SGD | 100.00% | 96.00% | 58.99% | 56.36% |
| | SAM | 100.00% | 96.48% | 61.81% | 54.01% |
| | AdvReg | 99.99% | 95.66% | 57.44% | 56.32% |
| | SELENA | 95.75% | 94.62% | 55.49% | 51.77% |
| | **SharpReg(1.5,0.01,0.1)** | 97.92% | 93.34% | 52.86% | 53.48% |
| Purchase-100 | SGD | 100.00% | 85.50% | 67.13% | 65.59% |
| | SAM | 100.00% | 85.54% | 67.54% | 66.06% |
| | MemGuard | 99.98% | 83.2% | 58.7% | 65.8% |
| | AdvReg | 94.80% | 78.94% | 59.07% | 59.16% |
| | SELENA | 88.08% | 81.24% | 54.37% | 54.39% |
| | **SharpReg(2.0,0.01,0.6)** | 98.78% | 82.29% | 61.16% | 61.27% |
| Texas-100 | SGD | 78.28% | 50.83% | 65.20% | 64.5% |
| | SAM | 81.17% | 51.34% | 66.59% | 65.36% |
| | MemGuard | 79.3% | 52.3% | 63.0% | 64.7% |
| | AdvReg | 73.60% | 49.44% | 63.45% | 63.63% |
| | SELENA | 60.24% | 52.40% | 54.84% | 54.86% |
| | **SharpReg(1.0,0.001,0.05)** | 64.65% | 49.49% | 59.89% | 58.51% |

Table 5: Privacy vs Generalization tradeoff for SAM and SGD using InceptionV4 and Resnet18

| Dataset | Model | Optimizer | Test Acc | Single-query Acc | Multi-query Acc |
|---------|-------|-----------|----------|------------------|-----------------|
| CIFAR-100 | Resnet18 | SGD | 78.42% | 74.31% | 71.51% |
|  |  | SAM | 78.74% | 77.45% | 68.50% |
|  | InceptionV4 | SGD | 77.44% | 77.22% | 71.17% |
|  |  | SAM | 79.60% | 80.82% | 67.73% |
| CIFAR-10 | Resnet18 | SGD | 95.18% | 57.90% | 57.79% |
|  |  | SAM | 96.16% | 60.05% | 55.37% |
|  | InceptionV4 | SGD | 94.26% | 61.60% | 58.24% |
|  |  | SAM | 95.76% | 64.41% | 55.83% |

value 0.1. The results are consistent with our findings that SAM tends to have higher test accuracy while having higher membership attack accuracy at the same time. Overall best attack accuracy is higher for SAM for all the cases although we find mixed findings for multi-query attack accuracy specifically.