# OpenReview forum: "On Memorization and Privacy Risks of Sharpness Aware Minimization"
_ICLR.cc/2024/Conference — ICLR 2024 Conference Withdrawn Submission_

### Official Review · Reviewer_Dp9T · 2023-10-22

**Soundness:** 2 fair
**Presentation:** 3 good
**Contribution:** 2 fair
**Rating:** 5
**Confidence:** 3

**Summary:**

This paper shows that sharpness aware minimization (SAM) tends to memorize more than SGD. This leads to concerns in data privacy. The authors further propose to seek 'sharp' minima to alleviate such privacy concerns. The proposed approach is observed to trade test accuracy for privacy against direct single query attacks and multi-query attacks.

**Strengths:**

S1. This paper evaluates sharpness/flatness through the lens of privacy and memorization. This perspective is novel.

S2. A new approach for alleviating privacy concerns regarding single/multi query attacks is approached based on the previous observation that flat solutions are more vulnerable. The proposed approach is simple -- seeking for sharp solutions.

**Weaknesses:**

W1. The toy example in Section 3.1 is not convincing enough. In this case, it so happens that the atypical samples in red class are overlapping with blue class. What if the atypical samples are away from the blue class? Moreover, the generalization and memorization of SAM can be a result of several intertwined factors (e.g., low rankness [1]), this toy example might be oversimplified such that some of factors are ignored.

W2. The numerical results in Tables 1 and 2 seem to not fully align with the claim on privacy. For example, on CIFAR-10 and CIFAR-100 datasets, SAM itself is helpful for label-only attack. This might suggest that the memorization of SAM can be more complicated.

W3.  The subgroup arguments in the introduction should be extended to elaborate more on the label noise experiment in SAM paper (Foret et al 2020). Because the label noise is randomly flipped in their experiments, typical and atypical samples are treated the same. And if SAM remembers the atypical data, does this suggest a dropping test accuracy (w.r.t. SGD) that is against the results in (Foret et al 2020)?


===== References =====

[1] https://arxiv.org/pdf/2305.16292.pdf

**Questions:**

See weekness.

---

### Official Review · Reviewer_3uQd · 2023-10-26

**Soundness:** 2 fair
**Presentation:** 1 poor
**Contribution:** 3 good
**Rating:** 3
**Confidence:** 4

**Summary:**

This work demonstrates that the accuracy benefits of Sharpness Aware Minimization (SAM) come at the tails of the data subpopulations, raising concerns for privacy. They empirically demonstrate this phenomenon over several experiments including vulnerability to Membership Attacks and accuracy over data buckets sorted by influence score.

**Strengths:**

The main contribution of the paper is strong in theory, demonstrating that the privacy risks of SAM. This claim is unintuitive and surprising, making for a good contribution. Moreover, the motivation of this paper is strong, and improving privacy is critical in deep learning.

Overall, I vote to reject this paper. However, if sufficient evidence is provided during this rebuttal, I'll gladly increase my score. I look forward to a productive discussion.

**Weaknesses:**

1. The paper has an incorrect claim in the contributions section. Namely, they mention they are the "first to empirically explore an explicit link between an optimization algorithm and privacy." This is completely incorrect and completely ignores the work of Differential Privacy. For example, Differentially Private Stochastic Gradient Descent is a well-known work that uses different optimization algorithms to achieve privacy (see "Stochastic gradient descent with differentially private updates"). Please remove this claim.
2. The related works completely miss all of the Differentially Private literature, which is very relevant. This needs to be added.
3. The notation is very difficult to understand and poorly done. I understand this paper borrows the notation from the Feldman paper, but a more detailed notation section is needed.
4. Why is $\mathcal{A}(D') = \mathcal{D} - (x_i, y_i)$, Do you mean set minus, i.e.$ D' = \mathcal{D} \setminus \{(x_i, y_i)\}$. Moreover, isn't the output of $\mathcal{A}$ a classifier, not a data set? This line needs revisioning.
5. You should use operatorname for operation names such as $\operatorname{mem}$.
6. The first sentence of section 2.1 is not in format; it should be $f(x; \theta)$.
7. The color choices in Figure 1 are poor. It is difficult to tell apart the different reds and point types.
8. The explanation around equation 3 is confusing. I thought $\mathcal{I}$ is defined for the classes. However, you mention you sort test data points by $\mathcal{I}$. What is the $\mathcal{I}$ of a single datapoint?
9. One of my main questions is whether it seems that, more than anything, test accuracy and single query attack accuracy are highly correlated. The two most convincing points of evidence that SAM is correlated with less privacy are in Table 1. However, is it not just the case up to noise that the model with the higher test accuracy almost always has the highest single-query accuracy? Then, SAM, in this manner, causes an increase in single query attack accuracy because it has higher test accuracy. This seems pretty consistent across all experiments. Moreover, it seems that using SharpReg causes less attack accuracy since it decreases test accuracy. What is the correlation between test accuracy and single query accuracy, and is the effect of SAM on accuracy not caused only by this increase in test accuracy? In my opinion, this needs much more development and is lacking in the paper.
10. Moreover, Figure 3a is used to claim that SAM's improvements come from points that require more memorization. However, Figure 3a could also be interpreted as SAM increasing accuracy on points with lower default accuracy. Is the correlation with the memorization level or the difficulty of the data point itself? Figure 3a does not distinguish between the two, which is critical for your paper.
11. No theoretical analysis or even intuitive explanation is given as to why this effect could happen. This is crucial, in my opinion, since the fundamental explanation of why SAM hurts privacy is missing.
12. Existing papers have analyzed SAM and its connection to privacy already. Namely, the paper "Differentially Private Sharpness-Aware Training" analyzes how flatness can actually boost privacy gains, which is the opposite of the claim in the paper. This paper does not cite that paper and explain the differences in claims at all. Although it is a recent paper, this paper was accepted and presented much before this paper was submitted, so I expect an explanation of why the claims run opposite to each other.
13. The explanation of what SharpReg is needs to come earlier.

**Questions:**

See above.

---

### Official Review · Reviewer_AV1Z · 2023-10-29

**Soundness:** 2 fair
**Presentation:** 2 fair
**Contribution:** 2 fair
**Rating:** 5
**Confidence:** 2

**Summary:**

- Influence entropy metric is introduced in order to understand the generalisation performance of a learning algorithm. This metric is used to discuss the generalisation gains of SAM , which are shown to be primarily for atypical samples.
- Higher privacy risk to Membership Inference Attacks is shown for SAM through extensive experiments.
- Sharper minima as privacy defense(SharpReg) is proposed and it's effectiveness is discussed for 4 datasets

**** Post-rebuttal period *****

No replies from the authors in the discussion period and the concerns raised in the weaknesses and question section remain unaddressed. Hence, Changing my score from borderline accept to borderline reject.

**Strengths:**

- A structured framework to discuss the generalisation gains of learning algorithms through the lens of memorization is proposed, which is used to show that the success of SAM is more pronounced for atypical examples.
- This observation is verified through higher membership inference attack risk for SAM
- Effectiveness of SharpReg as a privacy defense is demonstrated on multiple datasets

**Weaknesses:**

- Best results in all tables in the paper could be highlighted for more readability.

- Proposed entropy metric is computationally expensive to compute.

**** Additional Weaknesses ****

- There are some typos in important places as highlighted by other reviewers as well. In the memorization and influence score definition on Page 3, $h(x_i) - y_i$ should actually be $h(x_i) = y_i$ .

- Comparison of SharpReg with other privacy preserving methods is missing, also keeping in mind the influence metric proposed. Do other methods impact specific groups more ?

- The toy example presents an oversimplified picture of a real-world setting and does not help in getting intuition in the general case.

**Questions:**

> But as observed by Feldman & Zhang (2020), and as also confirmed by our privacy leaks experiments, these memorization and influence scores are largely a property of the data, rather than that of model architectures or other variations in the training.

- I'm not sure about the correctness of this statement. Learning is a property of the model architecture and the optimization algorithm [1].

- Why do you think SAM does not have higher privacy risk consistently for multi-query attacks ?(Table 1)

Additionally, please refer to weaknesses section for questions as well.




[1] Harshay Shah, Sung Min Park, Andrew Ilyas, and Aleksander Madry. “ModelDiff: A Framework for Comparing Learning Algorithms”. In: arXiv preprint arXiv:2211.12491.

---

### Official Review · Reviewer_gjgb · 2023-10-31

**Soundness:** 2 fair
**Presentation:** 2 fair
**Contribution:** 2 fair
**Rating:** 3
**Confidence:** 4

**Summary:**

A line of work in deep learning explores links between a neural network's test accuracy and the loss function in the neighborhood of the parameters. Empirically, we see that "flat" minima tend to generalize better. Another line of work investigates how neural network memorization might improve generalization performance. This paper aims to connect these topics, suggesting that flatter minima generalize better because they memorize more.

The main experiments run multiple "sharpness-aware" optimization algorithms and break down their performance across subsets of the data, where the subsets are grouped by a measure of memorization. Existing methods aim for flat minima; to test their hypotheses, the authors introduce an algorithm that searches for "sharper" minima.

Memorization is associated with privacy risks. A large section of the paper investigates how these different algorithms perform under membership inference attacks.

**Strengths:**

I think the paper raises a number of excellent questions. Understanding links between neural network loss landscapes and memorization is an interesting direction for research. As far as I am aware, prior work has not pursued these questions.

The experiments searching for sharp minima really add a lot to the paper.

**Weaknesses:**

I feel this paper fails to "connect the dots" to tell a compelling story. The experiments suggest that SAM and SWA have higher accuracy on points that experience higher memorization, but, as far as I can tell, we see no clear evidence that the models more strongly memorize these points. Perhaps the algorithms do better on these points for some other reason?

The toy example highlights this shortcoming. We see a data distribution (built from subpopulations) where SAM outperforms vanilla SGD. The paper says that "the gain in generalization could potentially come from those atypical data subgroups." Why include such an example if it is not absolutely clear that the gains come from memorizing atypical data?

The paper also has serious presentation issues that made it difficult to understand and draw conclusions. For this and the above reason, I advocate for rejection.

Key presentation issues include the following.
- The entropy definition (Eq 3) appears to have a serious typo. It seems to depend only on the numbers of points in each class. Because of this, I could not interpret the (central) results relating memorization to accuracy. Related but less important issues include the overloaded index $i$ in Eq 3 and bucket numbering (the caption of Fig 2 says bucket 5 has high memorization, while Fig 3 seems to say the opposite).
- As mentioned above, I did not understand what conclusions to draw from the toy example.
- I sometimes had trouble understanding what conclusions to draw from the results. I think Contributions (Sec 1.1) and Conclusion and Future Work (Sec 6) reflect this issue: the only concrete takeaway I identify in them is that SAM may come with additional privacy risks.

Although it did not factor heavily in my decision, I believe the paper has a flawed discussion around "learning" and "memorization." The paper equates learning with compression (page 1) and talks about a spectrum with "perfect learning on one end and perfect memorization on the other" (page 2). However, recent work (Feldman 2020, who they cite, as well as [1] and [2]) highlights how memorization is compatible with learning. This should not be too surprising: the 1-nearest-neighbor classifier both learns and memorizes. These works go further, showing settings where memorization is actually *required* for accurate learning. Putting these concepts on opposite ends of a spectrum requires using nonstandard definitions.

[1] Brown, Gavin, et al. "When is memorization of irrelevant training data necessary for high-accuracy learning?." Proceedings of the 53rd annual ACM SIGACT symposium on theory of computing. 2021.

[2] Cheng, Chen, John Duchi, and Rohith Kuditipudi. "Memorize to generalize: on the necessity of interpolation in high dimensional linear regression." Conference on Learning Theory. PMLR, 2022.

**Questions:**

Does Eq 3 contain an error?

The Contributions section says "To the best of our knowledge, our work is the first to empirically explore an explicit link between an optimization algorithm and privacy." What is meant by this sentence? As the related works section lays out, there is extensive prior empirical work on optimization algorithms and privacy (e.g., optimization algorithms that satisfy differential privacy).

Why are the lines for SGD and SAM different between Figures 3a and 4a?

Can we conclude that SAM's accuracy gain over SGD derives (at least in part) from stronger memorization of some examples?

If so, can you lay out the steps that get us there?

If not, can you suggest some experiments that might, in principle, allow us to draw such a conclusion?

---

### Official Review · Reviewer_mP5k · 2023-11-10

**Soundness:** 2 fair
**Presentation:** 3 good
**Contribution:** 2 fair
**Rating:** 5
**Confidence:** 3

**Summary:**

This paper studies memorization of shaprness aware minimization (SAM). By looking at the test accuracy of data in different buckets categorized by different memorization scores, the authors conclude that SAM and SWA (stochastic weighted averaging) have more memorization concerns than SGD. The authors further compare test accuracy (generalization) and memorization (member inference attack accuracy) and show the trade-off between test accuracy and member inference accuracy. Finally, a new regularizer/optimizer that tries to find sharp instead of flat minimizer is proposed and evaluated.


==== discussion phase =======
No response provided. Changed from borderline accept to borderline reject.

**Strengths:**

Generalization and memorization is a timely topic.

The empirical evidence and discussion of generalization and memorization trade-off is interesting.

**Weaknesses:**

I would appreciate the authors clarify what they consider memorization is, and further justify the metrics used (influence score and membership inference attack accuracy). Specifically, is it possible that the conclusion about SAM and SWA have stronger memorization concerns due to inaccurate/inefficient metrics instead of the nature of the method? And also discuss the connection between memorization and privacy if possible. Better generalization leads to privacy risk is indeed counter intuitive, like the authors mentioned.

The proposed method SharpReg has different trade-off in generalization and memorization: worse test accuracy, but better membership inference attack performance. As the author noted, ShaprReg does not seem to be better than the other methods. And I am wondering if the authors have considered methods that can provide theoretical guarantees, such as differential privacy.

**Questions:**

Maybe I missed it, could the authors specify the member inference attack method in table 1?